# A Mixed Approach for Multi-Scale Assessment of Land System Dynamics and Future Scenario Development on the Vaucluse Department (Southeastern France)

**Carla Scorsino * and Marta Debolini**

UMR 1114 INRAE-UAPV EMMAH, 84914 Avignon, France; marta.debolini@inrae.fr

* Correspondence: scorsino.carla@hotmail.fr

**Abstract:** The Mediterranean Basin is at the same time a region of stark social and ecological contrasts and a global biodiversity hotspot, where complex local evolving land use and land cover patterns compose the region's landscapes. In this context, we aimed for a specific case study of the southeast of France, to assess land and farming systems' dynamics, to identify their underlying drivers, and to propose possible shared future scenarios for local policies' implementation. We based our analyses on a mixed approach and operated at downscale from territorial to local scale. First, we applied a quantitative statistical approach for the Vaucluse department. Then, we identified a subzone of the study area and pursued a local analysis through a qualitative and participative approach based on stakeholders' knowledge. The study highlighted two main dynamics in land and farming systems that involve several changes. The first one is a process of land abandonment, strongly connected to a peri-urbanization process in some areas or to the loss of traditional farming systems in others. The second one is a process of specialization, at both territorial and farm levels, that corresponds to an intensification process and is linked to vineyards' expansion dynamic with a landscape homogenizing effect.

**Keywords:** Mediterranean; agricultural dynamics; farming system; typology; participatory approach; agricultural cadaster

## 1. Introduction

Mediterranean land systems are characterized by a strong spatial heterogeneity, mainly due to the variability in terms of climate and land morphology: On a same area, different land use and land cover can coexist together with different land management strategies and land use intensity [1]. The region of the Mediterranean Basin, besides experiencing comparable climatic conditions, presents similarities due to close historical socio-economic evolutions and a development largely based on agriculture and natural resources. In particular, Mediterranean farming systems have a high level of complexity, compared with the northern systems, because of the landscape mosaic and also because of morphological and climatic factors. Also, from the point of view of farm and territorial characteristics, some differences can be identified: Farms with different sizes can coexist, often with a close relationship between urban and rural. At the same time, many Mediterranean agricultural systems can be labor intensive and not always competitive on the global market. At the same time, this area is highly vulnerable in terms of impact to current climate change and in general global changes [2,3], facing a rapid population increase, which is expected to worsen over the next decades [4]. These processes result in a complex landscape mosaic which can be subject to various contrasted trajectories on a same space unit [5]. In general, fertile coastal Mediterranean plains have experienced over the last

decades a process of agricultural intensification, together with a population increase related to the urbanization [6], causing increasing conflicts on land planning [7]. Farming systems' patterns and their trajectories from traditional systems to more intensive ones present similarities in the different areas of the Mediterranean Basin [8] and have the particularity to still maintain traditional farming [9]. Moreover, these dynamics caused the fragmentation of the residual natural vegetation and habitats, determining a loss of biodiversity and of ecosystem services provision [10,11]. Various case studies have been developed over the last decades in order to understand local existing dynamics, their drivers, and the effect on agricultural production, local food systems, and on biodiversity, e.g., [12–14]. However, most of the research focused on the relation between urban and agricultural areas [15], or on agricultural abandonment related to urbanization [16], whereas a research gap can be identified on the analysis of intensification processes around the Mediterranean area [17].

Southeastern France is a highly productive area where high added value productions are predominant, such as vineyards, orchards, and horticulture. They usually coexist in the same region with strong touristic, urban, and industrial pressures. The Vaucluse department can be considered as a relevant example of a region where agricultural production still represents one of the main economic activities, but with a shift from more traditional and extensive farming systems to more intensive ones [18]. Land price in the department are especially high and diverse: The median price is more than twice the national one but with a factor of 10 between the minimum and maximum price of agricultural land, based on the data from the *Sociétés d'aménagement foncier et d'établissement rural* SAFER [19] and LegiFrance [20]. Previous analyses led at the Mediterranean Basin scale identified peri-urbanization as a main land system change in this area [21]. This aspect is particularly relevant if we consider that the area does not have cities of medium-large size, but that the largest urban center of the department, Avignon, is a small-medium sized city of 90,000 inhabitants. This characteristic is typical of various Mediterranean territories: There is a widespread urbanization trend also outside from the main centers, resulting in a diffused urban sprawl [22]. However, some local dynamics, such as specialization of labelled vineyards or the intensification of farming practices [18,23], cannot be captured by the large-scale land use or land cover change analysis, and they need investigation at a more detailed level to be identified. Different approaches can be identified for land system change analysis, mainly depending on the spatial scale of interest. Local dynamics cannot easily be detected through statistical and quantitative methods, mainly in a limited time span. Moreover, some local changes can affect management practices without a shift on the land use (market, technical, or environmental farmer strategies), and these kinds of dynamics are not detectable through the standard geographical methods. On the other side, these changes can strongly affect local actors and farmers, because they rely more on agricultural practices and land management, and they can affect primary production and productions costs. For these reasons, there is a need for mixed approaches and studies combining quantitative methods and qualitative ones directly involving local actors in the analysis of the dynamics and their underlined drivers, as a needed step for shared local policies.

In this paper, we aimed to assess land and farming systems' dynamics both at territorial and at local scale, to identify their underlying drivers, and to propose possible shared future scenarios for local policies' implementation. We based our analyses on a mixed approach. In particular, we applied a quantitative statistical approach for the territorial scale; then we identified a subzone of the whole study area where we analyzed local knowledge accessed both through stakeholder interviews and through a participatory approach developed to obtain a shared vision of the territory. This participatory approach attempted to identify possible future scenarios, both wished and expected, and to propose local actions. The description of those local dynamics and their drivers could improve the understanding of land and farm system trajectories observed in other areas within the Mediterranean Basin, especially since this contribution focuses on a mixed region with both inland plain and hilly area that are two Mediterranean landscapes less analyzed in term of land dynamics [17]. Thus, specific dynamics in the case study will be then confronted and contextualized within global Mediterranean evolutions.

In the following sections, we will present: The study area (Section 2.1), the methods applied for the quantitative and qualitative approaches (Sections 2.2 and 2.3), and the results obtained in the two parallel analysis (Sections 3.1 and 3.2). Then, we will discuss our methods and results on the fourth section.

## 2. Materials and Methods

In this study, we applied a two steps methodology. First, we carried out a statistical analysis of the agricultural census data on two dates, 2000 and 2010, for the whole Vaucluse department at farming scale. These data were chosen because they include the most comprehensive and detailed information about farming systems and their management. These two years are the most recent dates in which the agricultural census has been realized. A new census is currently ongoing for 2020 but it is not still available. Through this process, we obtained the main trajectories of land system dynamics at territorial scale. Then, we applied a more qualitative methodology at local scale in order to understand dynamics affecting farming system that are not necessarily identifiable through cartographic land use/land cover analysis. Indeed, it appears that some dynamics (such as vineyards expansion) were too localized to be described at the landscape scale through quantitative tools such as spatial analyses (that mostly reveals peri-urbanization process) (Fusco et al., 2019), and needed a more local study. At the same time, others consequent changes are not manifested by a change in land use: farming system refers to the same type of land use, but changes concern general farming practices (market, technical or environmental farmer strategies). This second stage was carried out on the sub-zone of the Comtat Venaissin, as representative of the whole department in terms of farming systems diversity and dynamics. This second step consisted in semi-structured interviews with various territorial stakeholders, with the main objective to characterize farming systems in the local area and to identify the changes in terms of land use or land management practices that affected them. Moreover, we carried out a participative workshop that applied the "Territory game" methodology [24–26]. The objective was to obtain an overseen of current territorial dynamics and to lead the stakeholders to develop a shared vision for the future of their territory, trying to identify possible actions to be implemented starting from the shared diagnosis and scenarios.

### 2.1. Study Area

The Vaucluse department (Provence Alpes Cote d'Azur—-PACA Region, France) is a typical Mediterranean region characterized by a strong presence of perennial crops, such as vineyards and orchards (currently around 60% of the total utilized agricultural area—-UAA). It is a very heterogeneous area of more than 3600 km$^2$ going from the Avignon plain to the Mont Ventoux (1912 m above the see level—-a.s.l.). Within the Vaucluse department, the French Comtat Venaissin (Figure 1) is bounded by three relief areas: Dentelles de Montmirail in the north, Mont Ventoux as its eastern extremity, and the beginning of Monts de Vaucluse in the southern part. This area includes 26 municipalities with a total of 93,518 inhabitants. The three main urban areas are Carpentras, Monteux, and Pernes-les-Fontaines, which are comprised of 50,926 inhabitants.

The region is constituted of mixed landscapes and combined peri-urban internal plain in its western part (Rhone valley plain) and hilly area in the north and east parts.

From the 1930s to the 1970s, agriculture in the French Comtat Venaissin experienced its golden age thanks to a specialization on market gardening production, for local consumption but also for exportation, supported by the progressive establishment of Carpentras' irrigation canal. Market gardening and bocage landscape (i.e., an agricultural landscape of irregular and small parcels delimited by hedgerows) increased in the irrigated plain that benefits from deep alluvial soil. At the same time, orchards, table grapes, and labelled wine production emerged in the dry areas of Carpentras canal upstream, hills, and calcareous plateau, designing a landscape where groves, vineyards, and terraced vineyards (one the more favorable slopes) alternate with seminatural areas of "*garigue*". Since then, market gardening remained a very relevant sector on the Comtat economy, but in the last decades it

has been weakened by international market competition, and the initiatives of quality improvement and promotion (organic agriculture, quality approach) were not sufficient to completely preserve the activity. Viticulture and French Protected Designation of Origin (AOC or AOP)-labelled areas consequently expanded, including in the plain, and most of the municipalities are concerned by at least one designation with the consequence of the specialization of some areas in wine production and the evolution of homogenous landscapes with large vineyards and clustered habitat.

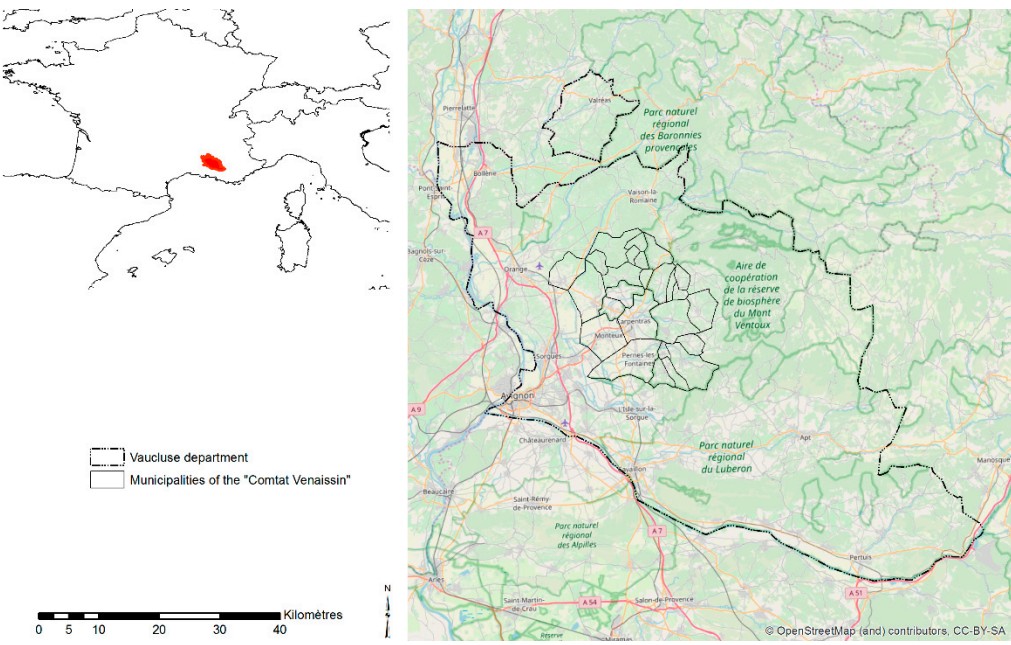

**Figure 1.** Localization of the study area.

## 2.2. Quantitative Assessment of Land System Dynamics at Territorial Scale

The applied methodology was structured on two main phases. First, we characterized the existing farming systems on the study area through principal component analysis and cluster analysis, in order to create some farms' classes considering the main land use and management indicators. The exploited database was the national agricultural census at individual farm level, which is an exhaustive and temporally extended base on farm management, structure and practices. From this database, we extract the farms belonging to the Vaucluse department, obtaining a whole sampling of 7723 farms in 2000 (around 123,000 ha) and 5850 farms in 2010 (around 113,430 ha). The considered variables for the farming system classification were: percentage of each cropping system on the farm, percentage of irrigated crops, farm dimensions (cultivated area, UAA), presence of quality labels, livestock quantity, farm work units, and farmers' age. All the variables were first of all analysed through a classical principal component analysis (PCA) in order to identify the main discriminant and relevant variables to be used for the cluster analysis. The PCA was then followed by a Hierarchical Classification on Principal Components (HCPC), which allows for unsupervised classification of individuals. This function combines main factors, hierarchical classification and partitioning to better visualize and emphasize similarities between individuals. All these algoritms were implement on R trhough the FactoMiner package. Then, we carried out an analysis of the recent trajectories on farming systems: the characterization of farming system has been realized for two time laps (2000 and 2010). This allowed tracing some farming system trajectories, identifying areas where different change processes took place, such as intensification or specialization of agricultural system. For this assessment, the farming system classes were aggregated at municipal level, as for privacy questions it was not possible to show them at farm scale.

*2.3. Qualitative Assessment of Land System Dynamics at Local Scale*

2.3.1. Semi-Structured Interview

A first set of data was collected through semi-structured interviews conducted in 2018 with 45 diversified local stakeholders. Those interviews had the main objective of characterizing farming systems in the area, and of understanding their recent dynamics and the underlined drivers. At the beginning of the interviews, we localized precisely the boundaries of the case study providing stakeholders a map: in addition to their understanding of the localization of the study area, the map allowed them to locate the spatial dynamics during the interview. Specific objectives were: (1) characterizing the different farming systems existing in the study area, with the final aim to build a farming system typology and compare it with the typology obtained from the statistical approach; (2) describing the evolutions that have affected those farming systems in the past 30 years and (3) identifying the drivers of those evolutions. Interviews were then structured in 3 phases. The interviewees were asked (i) to characterize the different farming systems coexisting in the area; (ii) to describe the different changes observed in those farming systems and identify the reasons of those changes; and finally (iii) to express their perception of the impacts of those changes for the future. In order to facilitate the conduct of the interviews, their report and their analysis, we built an interview guide that allowed to meet our objectives. We sought to interview different types of stakeholders having specific knowledge in the three main agricultural sectors of the study area: viticulture, fruit growing and market gardening. A total of 15 local stakeholders were interviewed: 5 from agricultural development institution, 4 from territorial authorities' administrations, 3 from the academic sector, and 3 from the agricultural associative sector. Moreover, we collected information from about 30 farmers from the three main type of production in Vaucluse (viticulture, market gardening and arboriculture). From the interviews' reports, we lead qualitative analysis by classifying data in tematic matrices. For example, we identified in the discourse what refer to a change in farming system or what refer to a driver of change. In order to build the farming typology we also achieved a comparative reading of the interviews.

2.3.2. The "Territory Game" to Build a Shared Vision of Current Dynamics and Future Perspectives

The second step of the local-scale methodology was based on a participative workshop, using the Territory Game methodology [24–26]. The Territory Game is a research-action-training approach. It is a participative and collective learning system that allows us to design, concretize, and evaluate a territorial project, by facilitating stakeholders to express their representations of their territory [26]. This methodology was based on a principle of both spatialization and consensus: It involves stakeholders to represent geographically territorial current dynamics and possible future evolutions, after they all agreed with them.

For the preparation of the workshop, we mobilized different types of data in order to provide the players background maps and cards with territorial diagnosis information (e.g., production under designation of origin, tourist attractiveness, etc.). The workshop took place in January 2019 and allowed us to describe current territorial dynamics and issues, with a focus on farming and food systems, and to propose future possible perspectives. The global question of the workshop was the following: "French Comtat Venaissin in 2050: Which achievable transitions towards resilient farm and food systems will guarantee the protection of biodiversity?"

The game was structured in three phases related to three specific objectives, with a collective oral report in between the two first steps and after the last one. First of all, stakeholders realized a spatialized territorial diagnosis mainly focused on current local dynamics. For that first step, we provided a background map of the territory and thematic cards gathering territorial diagnosis elements to complete or help mobilize stakeholders' knowledge. Then, the "players" were asked to develop a shared scenario responding to resiliency and biodiversity protection criteria (as a response to the global question of the workshop), and to produce a map that represents it. Finally, they had to formulate possible actions that could be easily and directly implemented in order to initiate new local dynamics according to the vision developed during the second stage. For this last step, the "players"

had to fill out a table for each action to specify the following aspects: What? Where? Who? How? In total, 17 participants attended the workshop, with a major representation of the associative sector: Nine participants were from the environmental or agricultural associative sector, three were farmers, three came from public institutions (regional council, agricultural chamber, research institute), and two from the private sector. One of the participant was also interviewed in the first step of data collection, and the three farmers had already been interviewed as part of another study. They were divided into three groups for discussion, each accompanied by a facilitator and an observer. On the Figure 2, we show a moment of the territory game in Avignon.

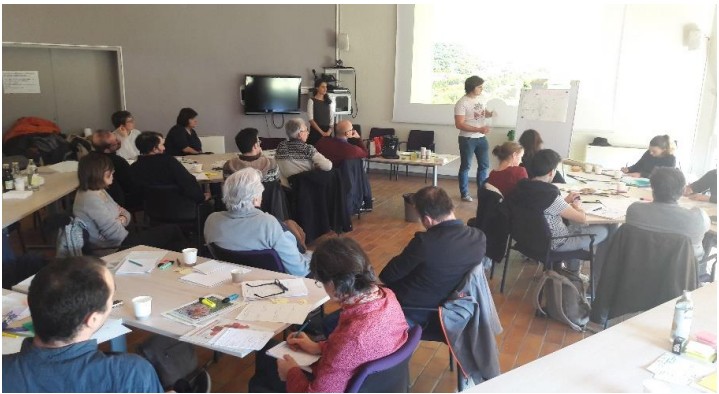

**Figure 2.** The Territory Game workshop during the restitution phase.

Qualitative analysis from the interviews and the participative workshop allowed us to confirm the changes observed with quantitative analysis, to complete it, and eventually to identify the drivers of those changes.

## 3. Results

In this section, we present the results obtained through the two groups of methods. First, we will show the results of the quantitative analysis and, in particular, we will present the farming system typology and its associated dynamics (Section 3.1). Then, we will present the results of the qualitative approaches and especially: the farming system typology (Section 3.2.1); the farming system dynamics and their drivers (Section 3.2.2); and the results of the territorial game (Sections 3.2.3 and 3.2.4).

### 3.1. Land and Farming System Dynamics in the Vaucluse Department

The principal component analysis - PCA highlighted the main discriminant variables to be used on the cluster analysis. In particular, the UAA and the percentage of each crop type were the most relevant, together with the surface irrigated, which were quite related to the percentage of vegetable crops. The first two dimensions of the PCA can explain around 25% of the total variability of the systems, which is not a low value, if we consider the large extension of the sample (more than 5000 farms in 2010 and more than 7000 in 2000).

From the cluster analysis, we obtained nine classes of farming systems in the study area: (1) Protected Designation of Origin (AOP) vineyards, (2) non-AOP vineyards, (3) aromatic crops' farms, (4) cereal farms, (5) fodder crops' and livestock farms, (6) grapes associated with orchards or small percentage of vineyards, (7) orchards, (8) vegetables farms, and (9) nursery farms. Table 1 shows the average values of some relevant variables estimated for each class of farming system for 2010. The main variable distinguishing the classes was the percentage of each crop on the farm, but other variables could give information about the intensity of the farming systems, such as the percentage of irrigated surfaces and the work units. The nine classes remained rather stable in the two observations (2000 and 2010), unless there was some tendency for concentration of the predominant crops for some farming systems.

**Table 1.** Main variables processed for farming system characterization.

| Class | UAA (ha) | Work Units | Livestock | Cereals (%) | Aromatic Crops (%) | Fodder Crops (%) | Vegetables (%) | AOP Vineyards (%) | Non AOP Vineyards (%) | Grapes (%) | Orchards (%) | Nursery (%) | Irrigation (%) |
|-------|----------|-----------|-----------|-------------|--------------------|------------------|----------------|-------------------|----------------------|------------|--------------|-------------|----------------|
| 1 | 18 | 2226 | 0 | 4 | 0 | 1 | 1 | 79 | 2 | 2 | 3 | 1 | 4 |
| 2 | 9 | 1040 | 0 | 1 | 0 | 0 | 1 | 19 | 66 | 5 | 4 | 0 | 4 |
| 3 | 44 | 1435 | 3 | 10 | 59 | 8 | 2 | 8 | 0 | 0 | 4 | 0 | 5 |
| 4 | 36 | 1528 | 1 | 81 | 0 | 2 | 7 | 2 | 0 | 0 | 3 | 1 | 12 |
| 5 | 41 | 1320 | 93 | 10 | 0 | 80 | 2 | 2 | 0 | 0 | 4 | 0 | 18 |
| 6 | 11 | 2055 | 0 | 1 | 0 | 0 | 3 | 18 | 1 | 53 | 16 | 0 | 36 |
| 7 | 11 | 2489 | 0 | 1 | 0 | 0 | 2 | 6 | 0 | 1 | 83 | 0 | 40 |
| 8 | 7 | 2623 | 0 | 3 | 0 | 0 | 75 | 1 | 0 | 0 | 3 | 0 | 72 |
| 9 | 4 | 3492 | 0 | 1 | 0 | 0 | 1 | 2 | 0 | 0 | 1 | 93 | 81 |

**Note**: UAA—-Utilised agricultural area; AOP—-Protected Designation of Origin.

After the characterization of the farming system, an aggregated analysis of the dynamics was carried out, in terms of total surface and of number of farms for each system in the whole department.

Figure 3 shows the total surfaces for each farming system in 2000 and 2010. The results showed a relevant increase on AOP vineyards and a decrease of the less specialized and rentable farming systems, such as cereal cultivation and table grapes usually cultivated in the mountain areas, which progressively reduced their surface. A specific dynamics observed in the study area was the increase of the number of farms and associated surface of the fodder and livestock farms. This corresponded to the new horse farms set up in the last few years in the region, associated to agro-tourism development. In some cases, i.e., for aromatic crops' farms and fodder crops' farms, the increase in terms of surface corresponded to a decrease in terms of farms' number, indicating the concentration of land tended to be on bigger farms. The changes in terms of farming systems were also aggregated at the municipal level in order to spatialize them. The resulting maps are shown on Figure 4. In the Vaucluse department, we can identify three main areas of changes: (1) In the northwestern part, there was a shift from a diversified mixed farming system to a wine-specialized one, (2) in the central part, there was a decrease of the specialized orchards' farms to more extensive/mixed ones, and (3) in the southern part, there was another area of specialization from mixed farming systems to specialized wine.

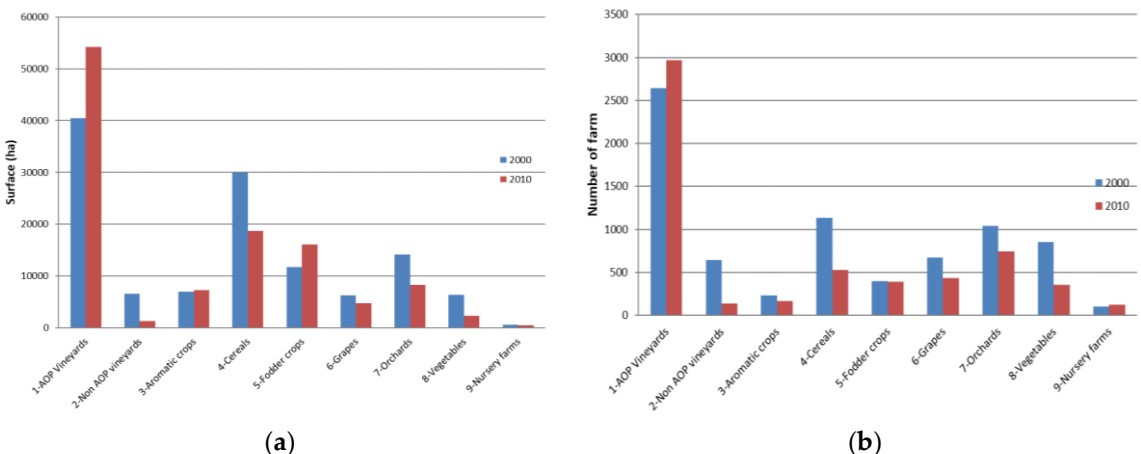

(**a**) (**b**)

**Figure 3.** Aggregated farming system changes from 2000 until 2010 for the whole Vaucluse department in terms of (**a**) surface and (**b**) number of farms.

The observed trajectories were classified into three main groups: (1) From non-AOP vineyards to AOP vineyards, which could be considered as a specialization of the farming system; (2) from cereals', orchards', or mixed farming systems to AOP vineyards, which in some cases could be considered as an intensification of the farming system; and (3) from cereal and orchards to fodder crops and livestock, which was an extensification of the farming systems. These dynamics are represented on Figure 5 in terms of percentage of change by municipality. We can observe that the intensification process was spread through the whole department, with a stronger intensity in the southern part, which was characterized by more mixed and extensive systems. The extensification trajectory was quite localized in the peri-urban area of Avignon and it was mainly due to the development of horse livestock systems. The specialization process was mainly present in the northern part of the department and related to the wine production.

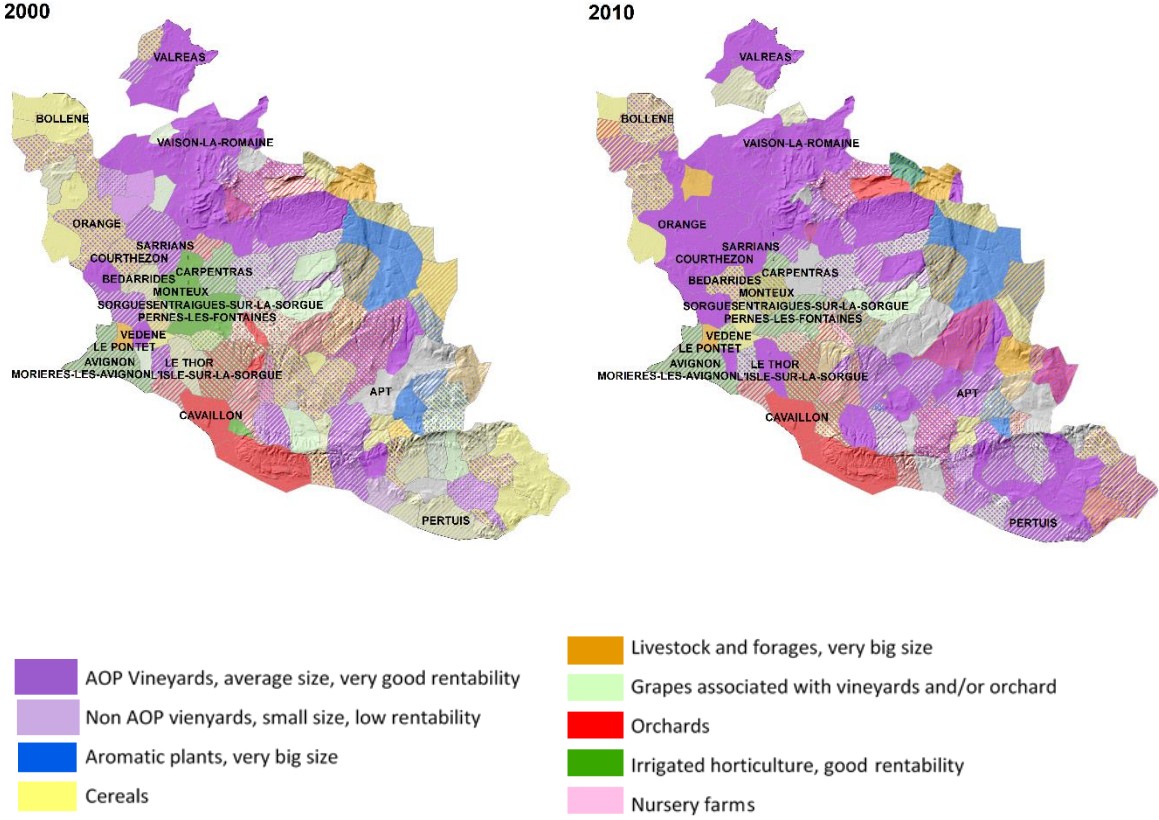

**Figure 4.** Farming systems' maps from 2000 and 2010. Municipalities with full color are those with only a prevalent farming system. Municipalities with linear hatching are those where two farming systems co-exist and, finally, municipalities with square cross-hatching are those where three or more farming systems co-exist.

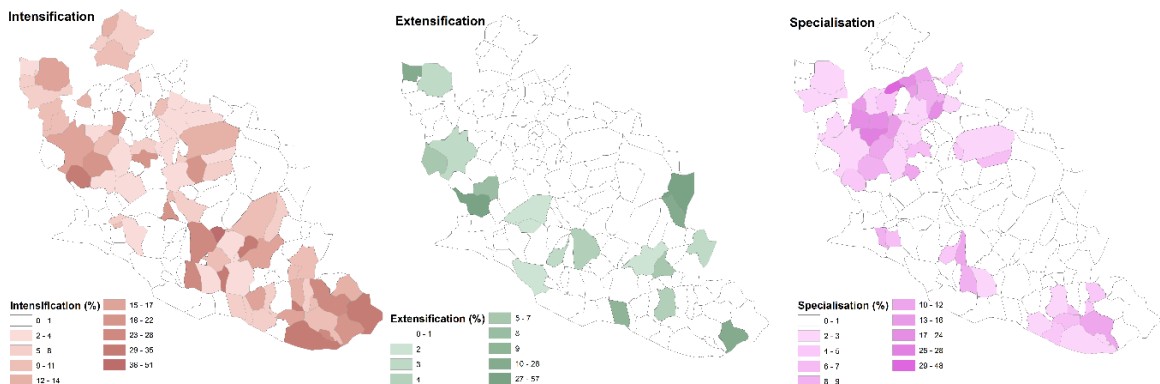

**Figure 5.** Intensification, extensification, and specialization trajectories on the Vaucluse department.

## 3.2. Farming System Dynamics at Local Scale

In order to better understand local dynamics and the underlined drivers, we focused our analysis on a sub-area of the department, for which we developed a work mainly based on local knowledge. We chose the central part of the department, indicated as the Comtat Venaissin, because it represented most of the farming systems of the whole department, and also most of the observed dynamics at territorial scale.

### 3.2.1. Farming System Typology

Based on stakeholders' knowledge, we built a typology to identify the main farming systems in the area (Figure 6), and the first criterion of definition was the main type of production. The typology

allowed identifying seven different farming systems: (1) Vineyards, without wine processing on the farm, that were generally part of a farmer cooperative. (2) Vineyards with wine making on the farm. (3) Orchards associated with vineyard: This is the typical and traditional "Ventoux farming system" combining table-grape, cherry, and wine-grape production. (4) Big and/or very specialized orchards that were specialized in one or two fruit productions and sell large volumes through long channels. It corresponded to fruit production systems that have generally enlarged or gotten very specialized in order to keep viable. (5) Diversified market gardening: It corresponded to farming systems that produce more than four kinds of vegetables within the year, and generally more than 10. (6) Specialized market gardening: It corresponded to farming systems that produce, in open-field and/or under nonheated tunnels, a maximum of four vegetables during the year, generally part of the main vegetables of the area (salad, strawberry, tomatoes, and melon). (7) Specialized market gardening with off-ground production: It is similar to the sixth type but part or all of the production was produce soil-free. This typology could be refined with two supplementary criteria: The production under organic certification or not for diversified market gardening systems, and the production under labelled designation of origin (French AOC or AOP) for vineyards.

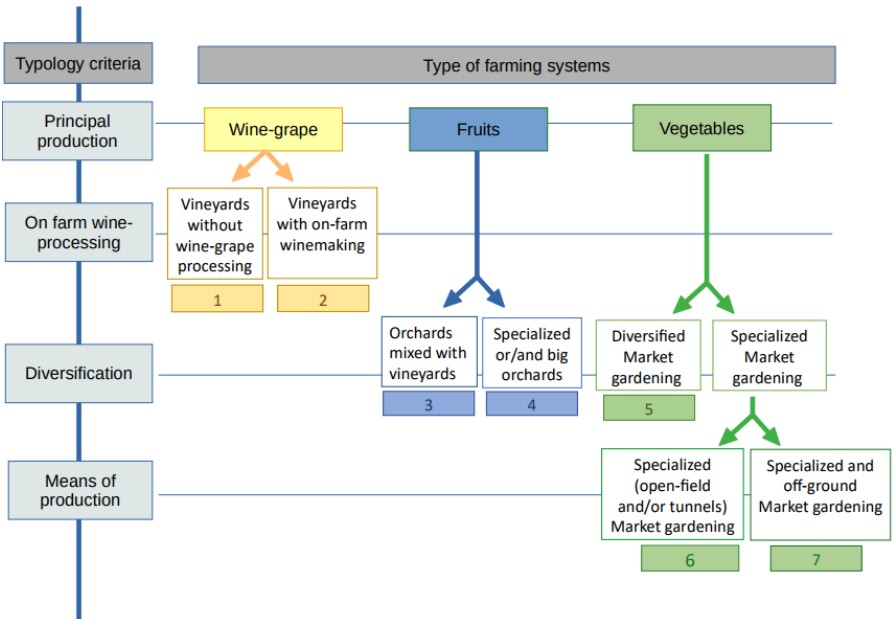

**Figure 6.** Farming system typology at local scale.

### 3.2.2. Farming System Dynamics and Their Drivers

Stakeholders identified different changes within the farm systems in the last 30 years (Table 2). Some trajectories involved different farming systems (general changes), while others just concerned a specific type of farming system. Apart from the conversion to vineyards, all the assessed dynamics affected only farming practices and did not lead to a change in the land use/land cover.

Farming systems in the area converted to organic agriculture for different reasons, including because of the changes affecting crop protection (removal of products, development of techniques) or valorization opportunities. If the conversion to organic farming concerns all farming systems, vineyards are the more concerned, because the conversion is easier compared to other productions, and, according to some stakeholders, because there was in the area a specific program, called "Ventoux Versant Bio", to promote it.

Orchards are globally getting bigger and more specialized, mostly because of phytosanitary pressures that requires big investments to be faced: The biggest farm resists those pressures by enlarging and specializing. The market gardening sector is experiencing two opposite trajectories. Indeed, while some farms get specialized in producing large volumes of few products (those that shippers

and wholesalers accept), others try to diversify their production, in particular to resist phytosanitary pressures (this generally goes hand in hand with the development of new commercialization channels). For what concerns wine production, most farmers settled in an area of designation chose to produce under designation of origin: This led to a homogenization in cultivated wine-grape species. The different trajectories of changes were generally followed because they allowed a better valorization of the production. Lots of changes in farming systems are driven by changes that happen at different levels of the global farming and food system, such as changes in the food sector (commercialization) or in the legislation for crop protection products. Other drivers are external to the farming and food system, such as phytosanitary pressures and tourist attractiveness. With those dynamics, stakeholders fear a homogenization and normalization of landscapes and farming systems within the territory and a decrease in the number of farmers and foresee the development and concentration of large-sized and intensive farms. They expect the continuation of the development of organic agriculture but with regulations that are more flexible.

**Table 2.** Farming system trajectories.

| Farming System Concerned | Trajectory | Reasons for Change/Drivers |
|---|---|---|
| General Change | Conversion to Organic Agriculture | Valorisation of the production; Changes in crop protection methods and product legislation; Incentives of public authorities; Market demand |
| General change | Conversion to vineyards | Valorisation of the production |
| General change | Development of short channel commercialization | Valorisation of the production; Foreign competition; Tourist attractiveness |
| Orchards | Specialization and/or farm size increase | Sanitary pressures; Food sector functioning |
| Market gardening | Specialization | Food sector functioning |
| | Diversification | Sanitary pressures; Commercialization channels |
| Vineyards | Development of production under designation of origin | Valorisation of the production |

### 3.2.3. Territorial Diagnosis

Territorial diagnosis (Figure 7) realized by the three teams of the Territory Game allowed us to identify four different zones in the study area that faced different issues:

(1) The "comtadine" plain (zone 1) is the market gardening area. A strong peri-urbanization process is ongoing around the main villages, with a consequence of land abandonment because of urban pressure and land speculation. Nevertheless, there is also a process of resistance to urbanization with peri-urban agriculture and the emergence of some patches of market gardening in the urban zones. Peri-urbanization and agricultural infrastructures lead to land artificialization.

(2) At the foot of Mont Ventoux, in the plateau area (zone 2), there is a mosaic of seminatural and agricultural land, which offers agricultural and landscape diversity, and where more traditional farming systems remain (orchards mixed with vineyards) but are threatened by vineyards' expansion. The area is also experiencing land abandonment, mostly for economic reasons and because of a loss of agricultural vitality and low profitability of crop production.

(3) At the north of the plain (zone 3), the area is specialised in wine production under designation of origin with homogenous landscapes and farm systems that correspond to large plain vineyards. Farming systems in this area are stable, except a slight process of enlargement of vineyards.

(4) The north and east part of the area (zone 4) correspond to the reliefs of Dentelles de Montmirail and Mont Ventoux, with mostly seminatural land but also some terraced vineyards and dispersed orchards. This area is not affected by urbanization.

Stakeholders' diagnoses highlighted vineyards' expansion from the plain to the plateau area and qualify it as a process of homogenization of landscapes and farming systems with an impact on biodiversity and cultural heritage (loss of terroir). They also insisted on the decrease of cultivated

agricultural areas, due to both an artificialization process (peri-urbanization) and land abandonment. Those two phenomena seem linked and land abandonment is driven by agricultural land price, urban pressure, and agricultural profitability (including crop profitability).

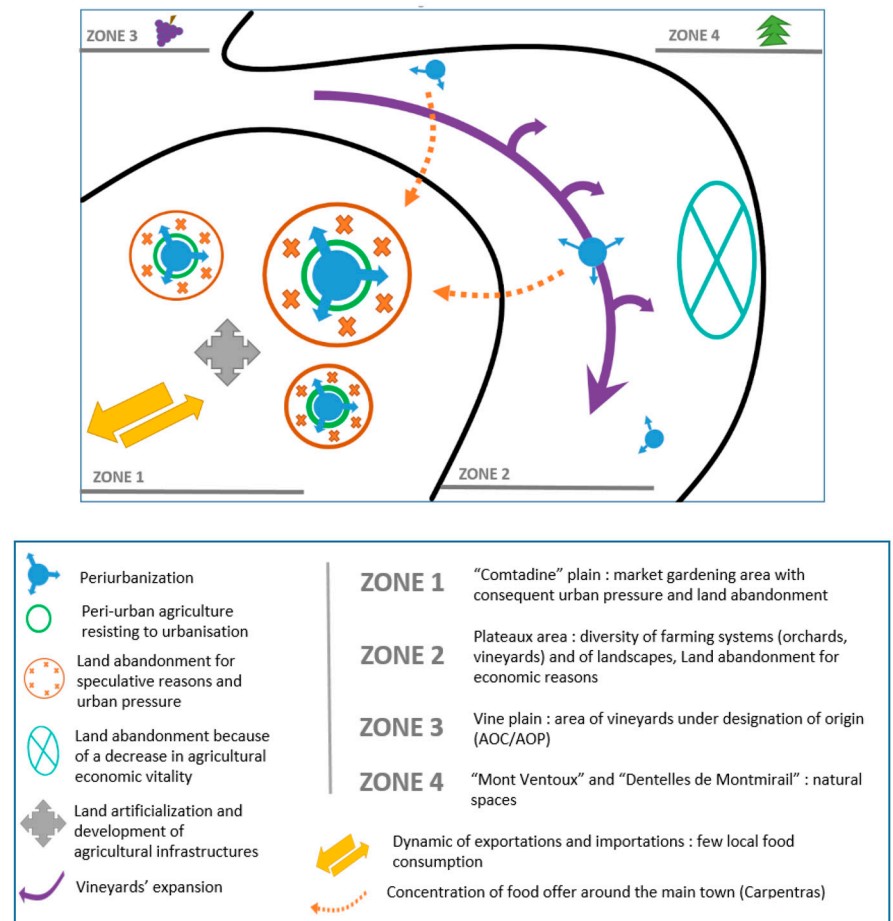

**Figure 7.** Synthetic and chromatic map of the diagnosis.

### 3.2.4. Shared Scenarios for Future Evolutions and Actions' Propositions

The vision that stakeholders imagined for their territory (Figure 8) and the propositions of action they formulated to achieve it were built around some general issues: (1) Improving agricultural, landscape, and biological diversity; (2) preserving and valuing agricultural land; (3) relocalizing food systems; 4) reinvigorating and preserving rural areas. In order to face territorial homogenization and specialization, stakeholders imagined a territory with stable vineyards for the benefit of a greater diversity of agricultural productions, also supported by the reintroduction of breeding in the area, by the preservation of traditional farming systems at the foot of Mont Ventoux, by the introduction of grain production in the plain, and by the valorization of unexploited agricultural areas through new productions (aromatic and medicinal plants, dry fruits, etc.). The desired agricultural development would promote a production that protects the environment and generates biodiversity, through the development of agroecology, agroforestry, and organic farming, and through the reinforcement of trees in landscape (restoration of hedges and the development of fruit trees' hedges), in particular, to adapt to pressure on the water resource.

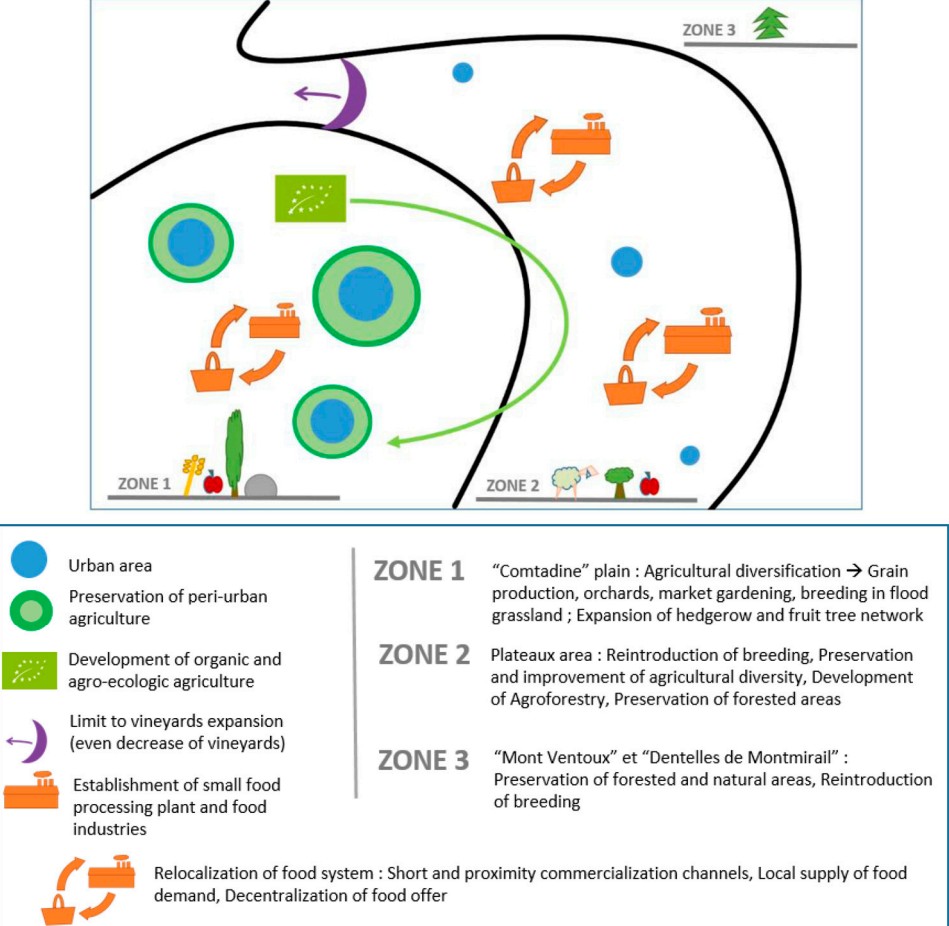

**Figure 8.** Synthetic and chromatic map of the scenario.

Besides this valorization of unexploited agricultural areas, various actions would address the global dynamic of land abandonment and the issues it raised, such as maintaining agricultural areas and facilitating agricultural land access for farmers. As an example of actions, stakeholders suggested for local authorities to buy agricultural lands and make them available for farmers. In peri-urban areas, the creation of green belt with peri-urban agriculture was proposed in order to preserve agricultural areas and to limit peri-urbanization and land artificialization. The revitalization of rural areas would also help in fighting land abandonment for economic reasons in the most rural areas.

Lastly, stakeholders aimed at relocalizing the food system and increasing the territorial food autonomy, thanks to actions such as the development of a food territorial project (*projets alimentaires territoriales*—-PAT), the development of local supply for collective catering, or the creation of a local food-processing plant, that have the common objective to reduce market-oriented agriculture for the benefit of local consumption. This global idea could also help to ensure a diversity of agricultural production in the area and to maintain agricultural activities thanks to high agricultural profitability (local and few intermediaries' valorization). Table 3 provides an overview of the stakeholders who may participate according to the different actions proposed by the players.

**Table 3.** Actors mobilized in actions proposed by the players.

| Actors | Preserve and Manage Agricultural Land | Assist Sustainable Agricultural Changes | Re-Localize Food System | Empower and Preserve Rural Areas |
|---|---|---|---|---|
| Local authorities | x | x | x | x |
| National and European authorities | – | x | x | – |
| Farmers | x | x | x | x |
| Rural and agricultural development institutions | – | x | x | x |
| Regional Natural Park | – | x | x | x |
| Land owners and managers | x | x | – | – |
| Food sector | – | x | x | – |
| Citizens | – | x | x | x |

## 4. Discussion and Conclusion

The study presented in this paper mobilized a mixed quantitative and qualitative methodology based on both statistical geographical approaches and territorial stakeholders' knowledge. This allowed us to understand land and farming system dynamics, both at territorial and local scales, and to have an overall vision of the underlined drivers. In terms of farming system typology, the two methods arrived at similar results, as in both cases the type of production was considered as the main discriminant factor. At the local case study, the second level of discrimination was more related to the processing, whereas at the landscape scale, variables linked to farm management, such as irrigation and work units, were considered. Diversification is a relevant factor in both typologies and, in fact, in the area we found at the same time very specialized and very diversified farming systems, contributing to a complex and heterogeneous landscape mosaic, typical of most Mediterranean regions. In terms of dynamics, both analyses, at landscape and local scales, highlighted the farming system specialization and simplification as one of the main trends, often coupled with urbanization.

Moreover, a possible shared scenario for the local case study was obtained together with some needed actions to set it up. In particular, this mixed approach allowed us to describe land and farming systems' dynamics that we could not identify with other approaches usually applied, such as land use and land cover change analyses, e.g., [17,27] or modelling approaches [28]. Indeed, the dynamics described that affected farming systems were mostly not associated with a change in land use, and some dynamics were too localized to be comprehended at a more global scale. The qualitative approach allowed us also to validate the farming system characterization obtained through the cluster analysis and to deepen it. The inclusion of the stakeholders' perception was also relevant to advance scenarios' development and overview of a possible future, and the local approach enabled us to access specificities and to deal with the realities of the territory.

Most dynamics observed in the case study were also observed at a global scale on the Mediteranean Basin [21,27], and a systematic review study [17] conducted at Mediterranean scale identified four main land dynamics that included three of the territorial dynamics stakeholders highlighted: Intensification, urbanization, and land abandonment.

Land abandonment is the main studied dynamic in the Mediterranean Basin [17] and this study highlighted multiple phenomenon of land abandonment that raised various issues pointed out by stakeholders, especially concerning farmer access to agricultural land, farmer renewal, or the decrease of cultivated area. In peri-urban areas, land abandonment is directly connected to the peri-urbanization process and engaged land price speculation and urbanization anticipation. In that case, land abandonment could be a precursor to urbanization and land artificialization. In fact, urbanization processes have been quite strong in the entire Mediteranean area for the past decade and have occurred at the expense of agricultural land [22]. However, the existence of small farming systems (market gardening) in the plain, could facilitate the persistence of agricultural land within urban sprawl [29] as it seems to be the case with peri-urban market gardening.

In more rural areas, land abandonment is driven by economic factors and due to a low agricultural vitality and profitability. In that case, land abandonment can lead to renaturalization or future artificialization or may correspond to the abandonment of traditional systems [17,30] and have variable impacts on biodiversity [31–33].

In this case study, land abandonment may also be related to the tourist attractiveness that impacts housing and land price, and foster speculative dynamics.

Dynamics that preserve the same type of land use (agricultural use) rely on two different processes as an answer to maintain agricultural profitability, and most of the identified trajectories were driven by economic factors. Firstly, a specialization dynamic is ongoing at both the territorial and farming system levels. Indeed, a homogenization of farming systems and landscapes is affecting the territory, due to an expansion of vineyards and wine production under designation of origin, impacting the territorial resiliency by increasing its vulnerability to viticulture crisis. In hilly areas, this dynamic is threatening traditional farming systems that remain and their associated landscapes: This loss of traditional landscapes is affecting other Mediterranean areas where economic factors drive agricultural dynamics [8] at the expense of cultural heritage. At both territorial and farm levels, specialization constitutes a form of intensification of the agricultural production that impacts agricultural biodiversity [27,34]. Then, other farming systems rely on increasing the sustainability of their practices or activating dynamics of conversion to organic or agroecological production, of diversification, and of short channels commercialization development.

All in all, those changes affecting farming systems can be gathered in two main dynamics that correspond to different strategies (to face current agricultural evolutions): Trajectories towards more sustainable farming practices (conversion to organic agriculture, diversification, mixed commercialization) or a process of specialization at both territorial and farm levels. Thus, within the same type of farming system, it is possible to observe opposite dynamics, such as specialization versus diversification, that reflect different farmers' strategies.

If some identified dynamics were in line with stakeholders' desired evolutions, others did not correspond to the vision local actors want to achieve. Stakeholders feel particularly concerned by land abandonment and artificialization processes and their issues. They formulated several propositions in order to improve the preservation and management of agricultural land, such as the acquisition of agricultural land by local authorities. They also had a special focus in preserving peri-urban agricultural land, which is a strong issue of the Mediterranean Basin [35], but not necessarily supported by local authorities as it is a source of conflict with housing and economic activities' area development [36].

The study revealed that stakeholders widely deplore the multi-scale specialization that is affecting the territory and threatening its resiliency. The global proposition made by stakeholders to relocalize the food system aims at limiting territorial specialization and fostering sustainable practices, but also at preserving agricultural activities. In this sense they frame several propositions of action, including the development of a Territoial Food Project (or PAT, Projet Alimentaire Territorial) that could bring a new type of territorial food governance [37]. PATs allow associating various stakeholders around food issues with a collective project that responds to economic, social, environmental, cultural, and sanitary challenges: It can constitute a tool for territorial public policy but the ability to be a sufficient lever to initiate an integrated local food policy is still debated [38,39].

Land system changes are the direct result of human decision making at different scales [28], including at a territorial level. Participative approaches, such as the Territory Game, allow engaging local stakeholders to put themselves in a prospective position, to think in terms of actions: It is an opportunity to formulate local development and land management recommendations, but also to devise potential territorial projects and dynamics of change. At the same time, as most of participatory approaches, this study can be affected by a bias due to the participation of actors particularly sensitives to the issues of territorial development and sustainable local agricultural production. In fact, most of the propositions coming from the shared scenarios go in this direction. Possible future development of the study could improve the relevance of the proposition, for instance presenting them during open

workshop on the territory and discussing them with local authorities and decision makers. Moreover, the analysis could be improved with more recent dynamics, considering the next agricultural census which will be released on the next months.

**Author Contributions:** Both authors equally contributed to this work in all its part. All authors have read and agreed to the published version of the manuscript.

**Funding:** This work was supported by the DIVECROP (Land system dynamics in the Mediterranean Basin across scales as relevant indicator for species diversity and local food systems, project ID: 10905) an ARIMNet2 project within an ERA-NET Action financed by the European Union under the Seventh Framework Programme for research, technological development, and demonstration. For the data access, the authors benefitted from the services of the "Centre d'accès sécurisé distant" (CASD), dedicated to authorized researchers pursuant to the permission from the "Comité français du Secret Statistique" (CSS).

**Acknowledgments:** The authors would like to acknowledge Fabrice Flamain for his fundamental work on the field surveys to the farmers. We thank all the persons that kindly accepted to be interviewed and to participate to the Territory Games.

**Conflicts of Interest:** The authors declare no conflict of interest.

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
