# Peer review of "A Mixed Approach for Multi-Scale Assessment of Land System Dynamics and Future Scenario Development on the Vaucluse Department (Southeastern France)"

_land, doi:10.3390/land9060180_

Round 1

Reviewer 1 Report

The submitted manuscript "A Mixed Approach for Multi-Scale Assessment of Land System Dynamics and Future Scenario Development on the Vaucluse Department (South-Eastern France)" concerns a very well-researched, mixed-method approach to investigate land- and farming-systems dynamics. I am willing to recommend it for publication, once a few issues – mostly minor ones – are addressed.

Introduction: The Section should cover a broader view of relevant land and farming systems concepts and trends in the Mediterranean area. A paper outline at the end of the Section would make the paper more readable.  

Methodology: Although they adopted very well-known statistical methods, the authors should illustrate the implementation of both principal component analysis and cluster analysis in a few lines by explaining, for instance, what algorithms and parameter settings they used. Moreover, the authors should explain how they analysed the interview transcripts or reports. Finally, the "Territory game" should be framed into the context of other participatory, deliberative, or scenario-building-support methods - for readers to appreciate the role of this method in the overall research design.  

Results: About the cluster analysis, a measure of variability should be provided along with average values. 

I couldn't find any result ensuing from the principal component analysis.

Table 1 is unclear, especially when compared to Figure 3: how have the classes been defined? (make sure the wording doesn't lay itself open to misinterpretation of the difference between classes and clusters). The variables in Table 1 seem to treated as Clusters in Figure 3.

The authors should either provide evidence (including that collected from the interviews) that the "Ventoux Versant Bio" programme had an effect, or reword the sentence in hypothetical terms.

Stakeholders' proposals, as expressed during the Territory game, read very mindful of farming system dynamics, but also quite standardised: they could apply to most peri-urban or rural areas in Europe. The authors may want to elaborate on this point, by checking whether it might be related to a strong influence they played in facilitating or in merely reporting the outcomes of the workshop or are somewhat indicative of some form of self-limitation or strict adherence to the expected conceptual framing of the problems on the side of the stakeholders. In either case, a reflection would be due to readers. Moreover, was there a consensus on all aspects of the scenario, or minority standpoints emerged as well?

Discussion and conclusion: The relation between cluster analysis and the qualitative methods may be defined better than by referring to validation. In general, a cross-cutting reflection on the results obtained from the application of the different methods could be deepened. The paper could close with the limitations of the study while envisaging the possible developments.  

Minor remarks

120: Legend keys and labels are hard to read

132: Invert abbreviation and in-full writing, and possibly introduce AOP as well

200: AOP has never been written in full before

239: in Figure 4, the hatched backgrounds are poorly readable, and the AOP key doesn't seem to match the colour in the map

256: in Figure 5, increase legend size because they are hard to read, and check consistency with text (extensification is not cited in the numbered list included in the paragraph above)

302: Table 2, a better layout would help sort general changes from specific ones

357: Why is Zone 3, as identified in the previous map (Fig. 7), left out Figure 8?

Reviewer 2 Report

Dear authors

Many thanks for allowing me to read this manuscript. Fascinating work!

Here are my comments that may enable you to strengthen your paper, especially your methodology and results:

Introduction

Provides a clear problem statement and introduces the aims and objectives of this study clearly.

Method

Described accurately. In 2.1) it is suggested to address the cultural values that characterised that landscape. Currently, the authors only address the productivity of the area, but any landscape is not only defined but by many other layers. It would benefit the reader to have a clearer idea what biophysical aspects are present as well as cultural and social values. 2.3) needs more clarity on how the data was analysed. Did the authors adopt an interpretative phenomenological approach? Were themes extracted from the transcribed interviews? How were the participatory workshops organised? What sort of questions were asked?

Results

This section needs a proper introduction. The reader jumps straight into the different sub-headings without understand the context, especially, on how the results are structured. At the end, it would benefit too to have a summary, as currently the results are too fragmented by the headings.

Discussion and Conclusion

Well-structured and clear.

All the best of luck

Reviewer 3 Report

The paper describes an important subject and clearly present the results of the research. 

A small drawback is using of quite outdated statistical data (2000, 2010) for statistical assessment of land dynamics (in this period 25% of farms vanished, which arise the question on farm number and UAA area in the current state), without any explanation on reasons staying behind this. However, it might be explained by data availability,  short sentence justifying why 2000 and 2010 were chosen for statistical analysis would clear this point. This is even more valid as interviews and experiments were conducted in 2018/2019, while the situation was probably different in 2010. The number of farms and UAA area under farming activities at the moment of running survey and territory game is not given. It is possible that the attitudes and beliefs of farmers, which could have been shaped by the changes that have taken place over the last years and were not presented in the article, influenced results of territory game and thus the conclusions.

Also, the aim of the paper was not fully realized. Authors presented dominating trends and potential trajectories, also formulated policy instruments and its impact, however, an issue of main drivers of those changes is not clearly presented. There are some drivers given in the paper as e.g. p3 ln.227 development of horse breeding is caused by the increase of agrotourism activities, but no specific methodology for the examination of those drivers is presented. This point is also not evaluated properly in conclusions. 

Detailed remarks:

p.5 ln.200/201 AOP vineyards/non-AOP vineyards - AOP is not defined/explained
